



# Technical Note: The Divide and Measure Nonconformity

Daniel Klotz[1], Martin Gauch[2], Frederik Kratzert[3], Grey Nearing[4], and Jakob Zscheischler[1,5]

[1]Department of Compound Environmental Risks, Helmholtz Centre for Environmental Research — UFZ, Leipzig, Germany
[2]Google Research, Zurich, Switzerland
[3]Google Research, Vienna, Austria
[4]Google Research, Mountain View, California, USA
[5]Technische Universität Dresden, Dresden, Germany

**Correspondence:** Daniel Klotz (daniel.klotz@ufz.de)

**Abstract.** The evaluation of model performance is an essential part of hydrological modeling. However, leveraging the full information that performance criteria provide, requires a deep understanding of their properties. This Technical Note focuses on a rather counterintuitive aspect of the perhaps most widely used hydrological metric, the Nash-Sutcliffe Efficiency (NSE). Specifically, we demonstrate that the overall NSE of a dataset is not bounded by the NSEs of all its partitions. We term

this phenomenon the "Divide and Measure Nonconformity". It follows naturally from the definition of the NSE, yet because modelers often subdivide datasets in a non-random way, the resulting behavior can have unintended consequences in practice. In this note we therefore discuss the implications of the "Divide and Measure Nonconformity", examine its empirical and theoretical properties, and provide recommendations for modelers to avoid drawing misleading conclusions.

## 1 Introduction

Measuring model performance is a foundational pillar of environmental modeling. For instance, in order to assure that a model is suited rainfall-runoff, we have to test how "good" it is. Over time, our community has established a set of performance criteria that cover different aspects of modelling. These criteria allow us to draw conclusions with regard to the evaluated model and should therefore exhibit consistent behaviour that follows our intuitions as modelers. However, when we use these criteria it is important to keep in mind that each one has specific properties — certain advantages and disadvantages — that are

relevant for interpreting results.

The Nash–Sutcliffe Efficiency (NSE; Nash and Sutcliffe, 1970) is the perhaps most used metrics in hydrology. In this contribution we show that the NSE exhibits a counterintuitive behavior (which, as far as we can tell, is so far undocumented), captured by the following exemplary anecdote. A hydrologist evaluates a model over a limited period of time and obtains an NSE value of, say, 0.77 (Fig. 1, blue partition). Then, a large event occurs and an isolated evaluation for that specific event

results in the slightly worse model performance of, say, 0.75 (Fig. 1, orange partition). One would then expect that the overall performance (i.e., a model evaluation over both the the blue and the orange partitions) should be bound by the values obtained during evaluation over each partition separately. However, the NSE over the entire time series in this example is 0.80 (Fig. 1, purple partition), which is higher than either partition.





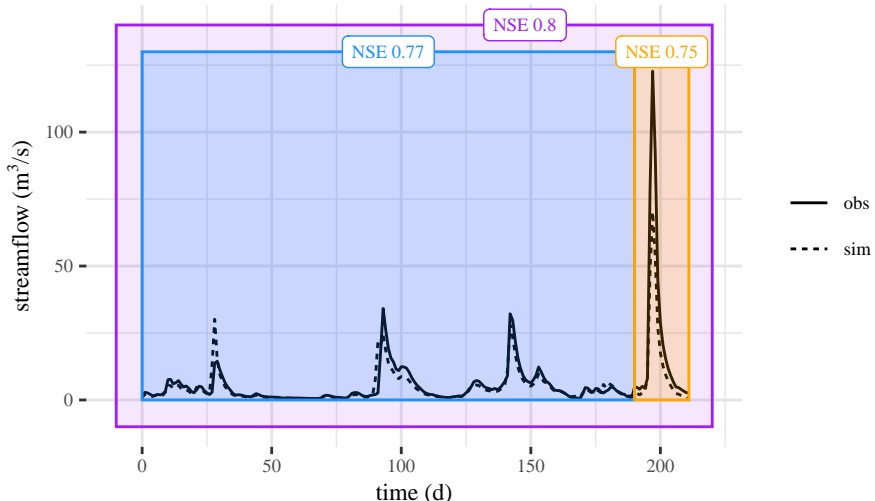

**Figure 1.** Example of the part–whole relationship within the Divide and Measure Nonconformity. The blue data partition has an NSE of 0.77, the orange data partition (that contains the peak event) has an NSE of 0.75. However, he overall NSE is 0.8 (violet partition), which is larger than both individual partitions.

We refer to the phenomenon that the overall NSE can be higher than the NSEs of data subdivisions as the *Divide and Measure Nonconformity* (DAMN). A natural question that follows from here is: What is the cause for the "DAMN behavior" in the example? To give an answer it is useful to consider the formal definition of the NSE:

$$\text{NSE} = 1 - \frac{\sum_{t=1}^{T}(o_t - s_t)^2}{\sum_{t=1}^{T}(o_t - \bar{o}_t)^2}, \tag{1}$$

where $o$ are observations, $s$ are simulations, $t$ is an index variable (usually assumed as time), $T$ is the overall number of time-steps the $NSE$ is computed over, and $\bar{o}$ is the average of the observations.

This is the standard definition of the NSE and it contains several different interpretations for the source of the "DAMN behaviour". One interpretation is that the new event shifted the mean of the observational data (which the NSE uses as a reference model for comparisons; Schaefli and Gupta, 2007) so that the observational mean became a worse estimate for the first partition (blue) as a portion of the superset (purple). Another way to explain this behavior is that the NSE gives very different results for partitions with different variability. The variance of the observations in the second (orange) partition is higher than the variance of observations over the superset (purple), meaning that the denominator in the NSE calculation is higher, resulting in a lower metric. One can imagine taking the squared error term (the numerator of the NSE metric) over only the second (orange) partition, but using the observational variance from the whole (purple) time period in the denominator, which would result in a value higher than the actual NSE value in the second period (orange) of 0.75.

The reflection from the previous paragraph concludes our motivational introduction. In what follows we provide a more in-depth exploration of the DAMN. We structure our exposition as follows: In the remainder of the introduction we discuss related





work (Sect. 1.1) and the relevance and potential implications of the DAMN (Sect. 1.2). Afterwards we conduct a case study where we empirically show that the overall NSE can only be equal or higher then the NSE values of all possible partitions (Sect. 2 and Sect. 3; Appendix B provides a corresponding theoretical treatment showing that this behavior logically follows from the definition of the NSE). In the last part of the manuscript we present our conclusion and provide some recommendations

for modellers that follow from our examination of the DAMN (Sect. 4).

## 1.1 Related work

The NSE is so important to hydrological modelling that there exist many publications that (critically) analyze its properties (e.g., Schaefli and Gupta, 2007; Mizukami et al., 2019; Clark et al., 2021; Gauch et al., 2023). Covering the full extent of the scientific discussion is out of scope for our Technical Note. Instead, we will just mention the few publications that are most

relevant for us: Gupta et al. (2009) use a decomposition of the NSE to show that the criterion favours models that provide conservative estimates of extremes. In contrast, our analysis provides a data-based view of how the NSE behaves when data is divided or combined. Clark et al. (2021) demonstrate inherent uncertainties of estimating the NSE and suggest using distributions of performance metrics to understand the inherent uncertainties. While their analysis focuses on the difficulties of finding a hypothetical "true NSE value", we focus on a specific behavior that concerns the part-whole relationship of the crite-

rion. Lastly, Duc and Sawada (2023) argue that the NSE is not necessarily well suited to compare rivers that exhibit different streamflow variances. This aligns with parts of our analysis, since we view the evaluation of multiply rivers as over multiply data partitions (the same logic as in our introductory example from Sect. 1 applies also here: Whether the mean of a time series is a better or worse estimator depends mainly on variance of the observations).

**Data splitting.** Data splitting is common practice in machine learning and data analysis. To our knowledge, the oldest records of data splitting go back in the early 20th century (Larson, 1931; Highleyman, 1962; Stone, 1974; Vapnik, 1991). These classical cases and the approaches that derived from them use random splitting. Thus the DAMN does not play an important role here (Sect. 1.2 provides an intuition for as to why). In hydrology, there exist two common situational (i.e., non-random) data splits: (1) the spatial data-split between catchments (e.g., Kratzert et al., 2019; Mai et al., 2022) and (2) the temporal data-split

for validating (for a recent discussion see Shen et al., 2022). Regarding (1), Feng et al. (2023) recently proposed an ad-hoc regional data partitioning for model evaluation. A perhaps more principled form of this technique can be found in the data-based splitting that have been put forward independently by Mayr et al. (2018) and Sweet et al. (2023). Both, on their own terms, propose to partition the data based on feature clusters. Either way, this type of informed (non-random) splitting is susceptible to the DAMN. Regarding (2), Klemeš (1986) introduced a-style of two-fold (cross-) validation to hydrology. Inter alia, he

proposed the so called *differential split sample test*. It is a type of non-random split that subdivides a hydrograph into parts that reflect specific hydrological processes — say, low flow and high flow periods. This type of splitting is common in hydrology, but since it is also an informed (non-random) splitting it is indeed exposed to the DAMN.

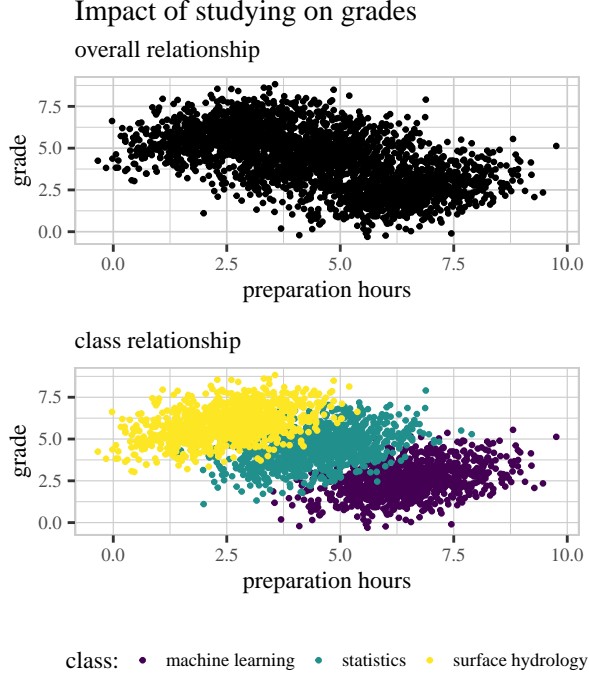

**Figure 2.** Toy example illustrating Simpson's Paradox, showing the relationship between the time spent studying and grades. Top: The "global" evaluation of the data suggests a negative effect of preparation time on the grade. Bottom: The "local" evaluation from splitting students by exam class shows positive correlation between study time and grades. Evaluators should account for both patterns — the global and the local — depending on the purpose of the analysis. Adapted from Wayland (2018).

**Statistical paradoxes.** Statisticians have coined many paradoxes. In particular, the DAMN is closely related to Simpson's
Paradox (Simpson, 1951; Wagner, 1982). Simpson's Paradox is an illustration how a possitive statistical associations can be
inverted under (non-random) data partitioning (Fig. 2). The DAMN can be seen as a special case of Simpson's Paradox, since
it describes the behavior of model performance metrics when (non-random) partitions of the data are combined (or, vice versa,
when the data is divided in partitions). Similarly, an amalgamation paradox (*sensu* Good and Mittal, 1987) can be seen as more
general form of Simpson's Paradox, describes how statistical association increase or decrease under different data combina-
tions. Hence, the DAMN can also be seen as a special case of an amalgamation paradox, where the measured performance
always can only increase when we combine data.





## 1.2 Relevance

**Settings.** There exist many examples of situations where modellers should be aware of the DAMN. For example: (1) Approaches that rely on sliding windows (e.g., Wagener et al., 2003). Here, one cannot derive the overall performance from the performances over different windows of the data, and NSE values calculated over sliding windows might appear smaller than the ones calculated over longer time periods of the same data. (2) Aggregating or comparing separate evaluation of different rivers — for example, as was done by Kratzert et al. (2019). For basins with low runoff variability, the mean is a better estimator than for basins with high variability. (3) Differential-split settings that divide the hydrograph into low flows and high flows (e.g., Klemeš, 1986). In this case, the low-flow NSE can be prone to having low values because the mean is a good estimator. However, high-flow NSEs will often suffer from larger overall errors.

**Generality.** In this Technical Note we restrict our discussion to the NSE because of its importance and ubiquity. However, the DAMN is a general phenomena and can occur for other performance criteria. For example, the Kling-Gupta Efficiency (KGE; Gupta et al., 2009) exhibits similar empirical behavior (we do not show this explicitly in this note, but encourage readers to explore it, e.g., by using our code, which provides an implementation of the KGE for testing). That said, simple average-based metrics such as MSE are not subject to the DAMN (see Appendix B2).

**Random partitions.** Albeit the DAMN can also occur with random subsets of the data (our theory applies also there; Appendix B) it is less of a concern in that then, since the independent sampling the overall NSE value should not deviate too much from the NSE values of the partitions. The intuition here follows the one given in our exemplary introduction (Sect. 1): In expectation, the means of two random partitions provide the same reference models.

**Limited sample size.** For model evaluation more data typically helps. This also holds true for situations where the DAMN is a concern, since the NSEs will behave less erratic when more data is used (see Clark et al., 2021). However, the DAMN as such is not a small-sample problem. It will occur whenever we divide the data into situations that have specific properties (e.g., when we divide the data along the temperature, while having a model that has a high predictive performance for low-temperature and low predictive performance for high temperatures). For example, the NSE remains susceptible to the DAMN independently of how well we are able to estimate the mean (or variance) of the data. That said, for the special case of time splits (for a given basin) it is indeed possible to argue that the occurrence of the DAMN is only due to limited data: If we had unlimited data for each partition, the inherent correlation structure (e.g., Shen et al., 2022) and the extreme value distribution of the streamflow (e.g., Clark et al., 2021) would not matter and our estimations of the mean (or variance) would converge to the same value for each partition — assuming no distribution shifts over time. Yet, sometimes we are interested in the performance of a model on subsets of the full available period, and for these cases no amount of overall available data will save us from the DAMN.





**Models with uncertainty predictions.** With access to models that do not (only) provide point predictions, but richer forms of prediction (say, interval or distributional predictions), modelers get access to more model performance criteria. Many of these criteria are evaluated first on a per sample level (e.g., by comparing the distributional estimation with the observed values) and then aggregated in a simple additive way (e.g., by taking the sum or the mean). Metrics based on proper scoring rules — such

as the Winkler Score (Winkler, 1972) for interval or the log-likelihood for distributional predictions — or metrics derived from information theoretical consideration (such as the cross-entropy) generally follow this scheme and are therefore typically not susceptible to the DAMN (see Appendix B).

## 2 Methods

We conduct two distinct experiments. The first experiment is purely a synthetic study. It examines how the overall NSE relates

to different NSE values of the partitions. The goal is to empirically show that the overall NSE can be higher (but not lower) than all of the individual NSE values of a partition. In the second experiment, we make a comparative analysis of the NSE and a derived "DAMN save" performance criterion. This experiment is based on real world data. Our goal is to examine the implications of the DAMN for a particular example. In the following subsections we explain both parts in detail.

### 2.1 Synthetic study

Our synthetic experiment demonstrates that the overall performance of a model (as measured by the NSE) is in many cases better than what all situational or data split performances would suggest. The setup is loosely inspired by Matejka and Fitzmaurice (2017): All data for the experiment derive from a single gauging station (namely, Priest Brook Near Winchendon (USGS ID #01162500), from Addor et al., 2017).

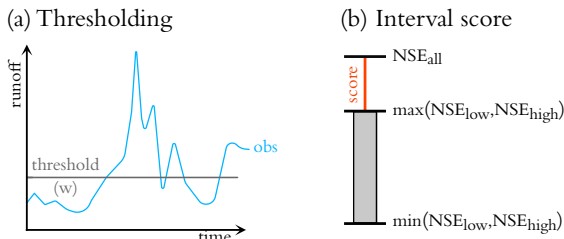

**Figure 3.** Exemplary depiction of the experimental setup. (a) For each model evaluation the data is split into two parts by a runoff threshold and three NSEs are computed: $NSE_{low}$ for data below the threshold, $NSE_{high}$ for data above the threshold, and $NSE_{all}$ for all data. (b) Then, the interval score is computed as the signed distance of $NSE_{all}$ from the interval between the $NSE_{low}$ and the $NSE_{high}$.

To generate simulations we (1) copied the streamflow observation data, (2) added noise to that observation data, (3) clipped

any resulting negative values to zero (to avoid streamflow that is trivially implausible), and (4) further optimize the resulting streamflow values themselves to reach a certain, prescribed NSE by using gradient descent. That is, we modify the data points





of the simulation (which in itself is just the observation with some noise) along the gradient given our loss function — and until the warranted performance (say, an NSE of 0.7) is reached. This allows us to build simulations that have defined NSE values for the data partitions. Specifically, we partition the observed streamflow into two parts: (1) "low-flows" that fall below

a threshold, and (2) "high-flows" that are at or above said threshold. We set the threshold using a desired fraction of data being designated as low- or high-flows. For example: $w = 0.2$ means the 20% smallest streamflow values are contained in the low-flow partition. We will refer to the NSE of the low-flows as $\text{NSE}_{\text{low}}$ and the NSE of the high-flows as $\text{NSE}_{\text{high}}$. We fix low-flow performance to $\text{NSE}_{\text{low}} = 0.5$ using the procedure outlined above (Appendix C2 provides similar results for $\text{NSE}_{\text{low}} = 0.25$ and $\text{NSE}_{\text{low}} = 0.75$). We vary both $w$ and $\text{NSE}_{\text{high}}$ between 0.1 and 0.9. For each point of the resulting grid we have three NSE

values: (1) $\text{NSE}_{\text{low}}$, (2) $\text{NSE}_{\text{high}}$, and (3) the overall $\text{NSE}_{\text{all}}$. We measure the practical effect of the DAMN using the signed distance of $\text{NSE}_{\text{all}}$ to the nearest edge of the NSEs of the partitions (either $\text{NSE}_{\text{low}}$ or $\text{NSE}_{\text{high}}$):

$$
I_s = \begin{cases} \text{NSE}_{\text{all}} - \text{NSE}_{\text{min}} & \text{if } \text{NSE}_{\text{all}} \leq \text{NSE}_{\text{min}} \\ 0 & \text{if } \text{NSE}_{\text{min}} < \text{NSE}_{\text{all}} < \text{NSE}_{\text{max}} \\ \text{NSE}_{\text{all}} - \text{NSE}_{\text{max}} & \text{if } \text{NSE}_{\text{all}} \geq \text{NSE}_{\text{max}} \end{cases} \tag{2}
$$

where $\text{NSE}_{\text{min}} = \min(\text{NSE}_{\text{low}}, \text{NSE}_{\text{high}})$ and $\text{NSE}_{\text{max}} = \max(\text{NSE}_{\text{low}}, \text{NSE}_{\text{high}})$ as shown in Fig. 3.

## 2.2    Comparative analysis

Our comparative analysis shows the influence of the DAMN by juxtaposing the behavior of the NSE with a derived performance criterion. This criterion is probably the simplest modification of the NSE that renders it "DAMN save". However, our intention with the new criterion is not to propose a new metric for hydrologists (even if it could be used as such). Rather, we want to introduce the criterion as a *tool for thought* to reason about the DAMN.

The most straightforward NSE modification we found is to use a fixed reference partition for the denominator of the NSE.

That is, instead of re-estimating the observational mean within the NSE for each (new) partition, we first choose a reference split and then compute the estimated variance from it (we also explored other, more complex modifications, but found them to be less insightful. Appendix A provides an example of such an exploration). Given the simple and straightforward nature of the modification, we refer to the "new" performance criterion as Low Effort NSE (LENSE):

$$
\text{LENSE} = 1 - \frac{\frac{1}{T}\sum_{t=1}^{T}(o_t - s_t)^2}{\frac{1}{T_{\text{R}}}\sum_{t=1}^{T_{\text{R}}}(o_t - \tilde{o}_{\text{R}})^2}, \tag{3}
$$

where $t$ is the sample index (which can but does not necessarily have to be a time index), $\bar{o}_{\text{R}}$ is the mean of the observations from a to-be-chosen reference partition, $T$ are the total number of timesteps in the evaluated partition, and $T_{\text{R}}$ are the number of timestep in the reference partition. In a certain sense, both, $T$ and $T_{\text{R}}$, are a result of the modification, since the different partitions for computing the errors and the observational variance make it so that the fractions to not necessarily reduce.

The LENSE follows a straightforward design principle: We use a reference set that is independent of the partition to trans-

form the right-hand side of NSE into a special case of a weighted mean squared error. This principle makes the LENSE





"DAMN save" because the denominator does re-normalize the squared error for each partition separately (Appendix B3 and Appendix B4 provide the corresponding formal proofs for the weighted mean squared error and the LENSE respectively).

The choice of the reference partition largely determines its interpretation. If, for example, the mean is supposed to be an estimate for the (true) mean of an underlying distribution (like, for example, in Schaefli and Gupta, 2007), then we should use as much data as possible to estimate it. In this case, it would be logical to use all data for the estimation — i.e.: training (in hydrology we refer to this partition as the calibration set), validation (in hydrology this partition typically does not exist or is subsumed into the calibration set), and test (in hydrology we refer to this partition as the validation set). If, on the other hand, we interpret the mean as a baseline model (like, for example, in Knoben et al., 2019), then it makes sense to use just the data that was used for model selection also for the estimation of the mean. One could also use the test split as a reference and recreate the NSE (the crucial difference is then that it is not allowed to update the reference split if new data arrives). Since the most convenient choice for such a reference split is the training (calibration) split, we propose to use it for the canonical application of LENSE (also, this split remains unchanged when new data arrives for the model to be used in the future).

The LENSE is robust against the DAMN by design. Thus measuring its interval score with our our synthetic setup will yield zero values everywhere. We did indeed try this as a check, but do not show these results explicitly since very little information is provided (we nevertheless encourage interested readers to explore this by using the code we provide). However, it is still insightful to compare how the LENSE and the NSE behave. Specifically, we explore two aspects. To that end we use the model and real-world data from Kratzert et al. (2019). First, we show how the performance criteria compare when we evaluate them for the 531 basins from Kratzert et al. (2019). Here, we evaluate NSE as in Kratzert et al. (2019) and use the training period as reference partition for LENSE. Second, we inspect the overall performance according to the NSE and LENSE related to the corresponding performances of different hydrological years for an arid catchment. We specifically chose an arid catchment here, since the mean of the runoff varies there more considerably between individual hydrological years. As before, we use the training period as reference partition for the LENSE.

For both parts of the comparative analysis we use the ensemble Long Short-Term Memory network from Kratzert et al. (2019) as hydrological models, but note that the model choice is not of importance (for comparison, see Appendix C provides some example cumulative distribution functions for other models).

## 3 Results and discussion

### 3.1 Synthetic study

Based on our synthetic experiment we find that $\text{NSE}_{\text{all}}$ can be outside of the range of the the NSEs spanned by the partitions (Fig. 4). Furthermore, the absence of negative interval scores indicates that the lowest-valued NSE of all partitions is a lower bound for $\text{NSE}_{\text{all}}$, which we confirm with theoretical considerations (Appendix B). Similarly, the existence of positive interval scores indicates that there is no trivial upper bound for the $\text{NSE}_{\text{all}}$ below its maximum of 1. We can also see that the interval scores tend to be highest when the NSEs of the partitions are equal, that is, $\text{NSE}_{\text{high}} = \text{NSE}_{\text{low}} = 0.5$. Intuitively from a statistical perspective, this makes sense: this is where the interval is the thinnest — and due to the lower bound, the $\text{NSE}_{\text{all}}$ can

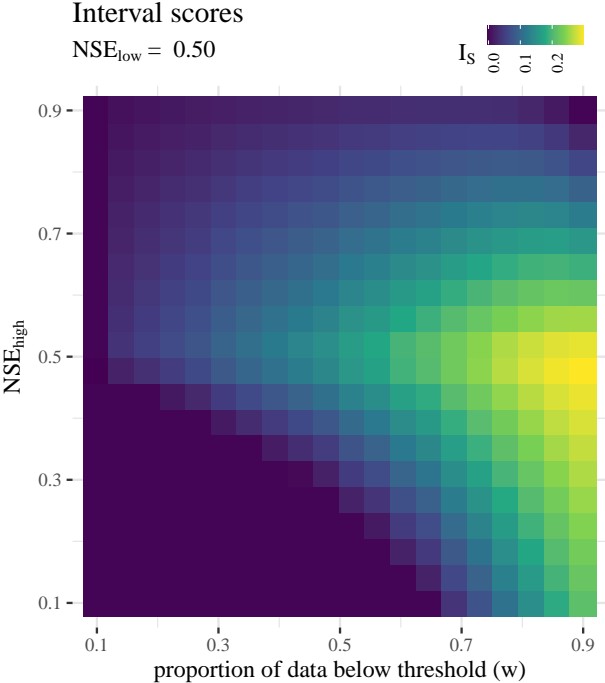

**Figure 4.** Interval scores $I_s$ as defined by Eq. (2). $NSE_{low}$ is set to the value 0.5. $NSE_{high}$ (y-axis) and the fraction $w$ of data in the lower partition (x-axis) are varied between 0.1 and 0.9.

only be above or exactly equal. Interestingly though, the highest interval score is only reached with the largest lower partition
we considered (90% of the data). Here, we do not only have the thin interval, but this is also the situation where we would
expect that the mean of the high-flow data is the furthest from the mean of the low-flows (since the mean of the low-flows does
not change much with the additional high-flows, while the highest high-flows have a substantially higher mean than the lower
ones). Thus, when we introduce the high-flow data into the $NSE_{all}$ computation it yields the largest difference.

### 3.2 Comparative analysis: NSE and LENSE

The comparison of the NSE and the LENSE for the LSTM ensemble and the 531 basins from Kratzert et al. (2019) shows that
the LENSE tends to yield lower values than the NSE, except for the best performing basins (Fig. 5). There, the LENSE values
are slightly higher than the NSE values. However, since the performance on these basins is already very close to the theoretical
best value (which is 1 for both criteria) the differences there are tiny.

For the yearly evaluation on an individual basin the NSE can vary substantially (Fig. 6). We note first that the LENSE
exhibits less variations over the years than the NSE. Further, we can see the overall LENSE is nicely enclosed within the
the values from the individual years — while the overall NSE is not. For four years the NSE values fall below 0.0, and for
two of the four they are below $-0.5$. These values are of particular interest, because the overall NSE is above 0.7. A naive



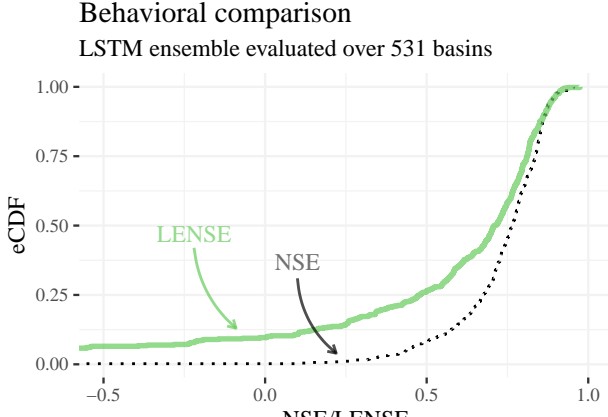

**Figure 5.** Empirical cumulative distribution functions of the NSE (black, dotted line) and the LENSE (green line) for the 531 CAMELS basins and the LSTM ensemble from Kratzert et al. (2019).

interpretation would suggest that the model degrades in performance in these years. However, a comparison to the respective LENSE values indicates that what we see here is largely an effect of the DAMN.

Another interesting phenomenon is that the NSE values from three hydrological years are higher than the corresponding LENSE values. Especially interesting here is that the worst LENSE values (-0.5) corresponds to an NSE that is above 0.0, which is far away from the supposed worst performance in terms of the NSE. This suggest that this year had a relatively high streamflow variance, with a relatively bad simulation.

To conclude, we re-emphasize that the purpose of the LENSE is not to propose a new metric or to replace the NSE. The

performance values we gain from LENSE should not be considered "more true" than those of the NSE. Rather, they show different aspects of the model behavior that are in the data, but easily overlooked if one only focuses on the NSE alone.

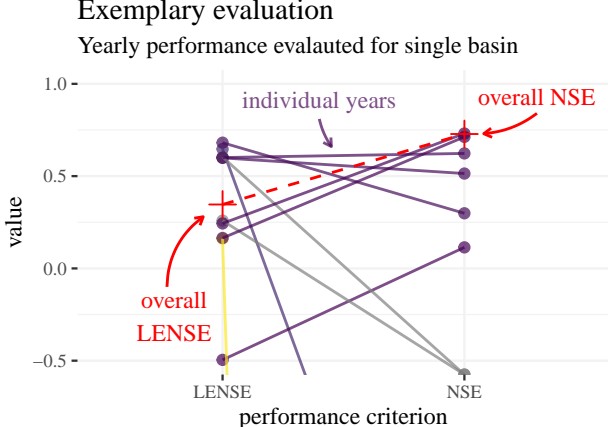

**Figure 6.** Comparison of NSE and LENSE in an arid basin. The colored dots show the performance for different hydrological years in the validation period; the crosses show the respective performances for the entire validation period. The large downward variability of NSE values exists because for some years the mean becomes an extremely good estimate for the daily runoff within certain periods. The LENSE, on the other hand, does not recompute the mean in the denominator for each validation year and has a stable estimation of the observational variance (see Eq. (3)). It is therefore more stable and less susceptible to such outlier years.

## 4 Conclusions

This contribution examines a part-whole relation that we coin "Divide and Measure Nonconformity" (DAMN). Specifically, the DAMN describes the phenomenon that the NSE of all the data can be higher than all the NSEs of subsets that together comprise the full dataset. That is, the global NSE can show counter-intuitive behaviour by not being bounded by the NSE values in all it's subsets. From a statistical point of view, the DAMN can therefore be seen as a sort of amalgamation paradox (Good and Mittal, 1987); and despite its counterintuitive appearance, the behavior can be well explained. Our goal with this Technical Note is not to eliminate the DAMN, but rather to make modellers more aware of it, explain how it manifests itself, and provide tools to check and think about it.

If we study model behavior in specific situations, we need to be aware of the DAMN. As an example, take an evaluation exercise where the model reaches very low (even negative) NSE values for an arid catchment, while obtaining high NSE values for a catchment with pronounced seasonality. Our analysis suggest that in this case the relative performance does not necessarily suggest model failure, but could also be related to the DAMN, since the mean is a very strong baseline for the arid catchment (which induces erratic NSE behavior).

Albeit our discussion revolves almost exclusively around NSE, many performance criteria are "DAMN susceptible". As demonstrated by our introduction of LENSE (a pseudo-performance criterion that serves as a thinking tool in our discussion), the strength of the effect depends mainly on the design of a given criterion. If a performance criterion is prone to the DAMN it implies that we cannot infer the global performance from looking at local performances.





With regard to follow-up work, we believe that our experimental setup suggests an interesting avenue for inquiry, which

we shall call "NSE kinetics". That is, to study how easy it is to improve or worsen the NSE by changing the observations or simulations with a given budget or constraints. For example, it might be easy to improve (worsen) the performance for basins where a model is weak by randomly improving some time points (by just adding noise). However, if one wants to improve (worsen) the simulation for a basin with pronounced seasonality and large amounts of high-quality data it might require a larger budget and changes to specific events. So far, scientists have studied the sensitivity and uncertainty of the NSE (e.g., Wright

et al., 2015; Clark et al., 2021, respectively). However, as far as we know, no one has yet examined a principled approach that is able to quantify the ease of change with respect to a given direction.

We conclude with the observation that the existence of phenomena like the DAMN underlines the importance of evaluating models with a range of different metrics — preferably tailored to the specific application at hand (Gauch et al., 2023). On top of that, we would like to push the community (and ourselves) to also always evaluate models with regard to the predictive uncer-

tainty when doing model comparisons and benchmarking exercises (e.g., Nearing et al., 2016, 2018; Mai et al., 2022; Beven, 2023). Typically, this will result in an additional workload for modellers, since it often means that a method for providing uncertainty estimates needs to be built (on top of a hydrological model that gives point predictions). However, existing uncertainty performance criteria not only provide additional information, but also are largely robust against the DAMN. Further, uncertainty plays an important role for hydrological predictions and should thus be included in our benchmarking efforts.

*Code and data availability.* We will make the code and data for the experiments and data of all produced results available online. The code for the experiments can be found at https://github.com/danklotz/a-damn-paper/tree/main. The hydrological simulations are based on the data from Kratzert et al. (2019) and based open source Python package NeuralHydrology (Kratzert et al., 2022). The streamflow that we used are from the publicly available CAMELS dataset by Newman et al. (2015) and Addor et al. (2017).





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





**Appendix A: Exploring a situation-equitable Nash–Sutcliffe Efficiency**

A modified NSE could also evaluate each sample differently and evaluate situations that are easy to predict more strictly and situations that are difficult to predict less strictly. We refer such a modification the Situation-Equitable NSE (SENSE). A
specific implementation that that uses the design principles from Sect. 2.2 is given by

$$\text{SENSE} = 1 - \frac{\sum_{t=1}^{T} \frac{1}{(o_t - \hat{\mu}_t)^2}(o_t - s_t)^2}{\sum_{t=1}^{T}(o_t - \hat{\mu}_t)^2}. \tag{A1}$$

Here, $\hat{\mu}_t^2$ is an estimation for the observational variance at time $t$, which we estimate by using a nearest neighbor approach that draws from a reference set:

$$\hat{\mu}_t = \frac{1}{K}\sum_{k=1}^{K} \text{kNN}(\boldsymbol{c}_t, \text{C}_{\text{R}}, k), \tag{A2}$$

where $\boldsymbol{c}_t = [o_{t-9}, o_{t-8} \ldots, o_{t-1}, o_t]$ is a situation vector containing the current observation and additional context in the form of preceding runoff values of the last 10 day, $\text{bmC}_{\text{R}} = \{\boldsymbol{c}_k, \boldsymbol{c}_{k-1}, \ldots, \boldsymbol{c}_0\}$ is a reference storage, and kNN yields the last observation $o_k$ within $\hat{\boldsymbol{c}}_k = [o_{k-9}, o_{k-8}, \ldots, o_{k-1}, o_k]$, which in itself is the k-th nearest neighbor of $\boldsymbol{c}_t$ within $\text{bmC}_{\text{R}}$. That is, we try to weight each given timestep $t$ by finding an approximation to the conditional variance of said timestep by taking the runoff observations in $\boldsymbol{c}_t$ and using the kNN regressor to find the $k$ most similar runoff vectors. From these we then derive a
situational estimation of the variance. Thus, the neighborhood of the kNN algorithm serves as the locality and its mean is our estimator for the expectation.

As a metric, SENSE would introduce two hyper-parameters: $k$ and $C$. It would also be possible to extend it and allow for arbitrary measures of similarity. Our analysis of the relative importance of choice $k$ shows that SENSE that the parameter is not particularly sensitive (Fig. A1). If one wants to develop more sophisticated extensions one could use more nuanced similarity
measures and also include explanatory variables (e.g., meteorological forcings) to better localize semantically similar samples (e.g., only match similar flow values in a rising limb).

**Appendix B: Connections between global and local performance criteria**

In the following, we consider a dataset $D_T = \{(\boldsymbol{o}_t, \boldsymbol{s}_t) \mid t \in 1, 2, \ldots, T\}$ with tuples of observations $\boldsymbol{o}_t \in \mathbb{R}^m$ and simulation values $\boldsymbol{s}_t \in \mathbb{R}^m$ that correspond to the index $t$ (which can, but not necessarily has to, correspond to a time step). In general $\boldsymbol{o}_t$
and $\boldsymbol{s}_t$ are vectors, but, for the sake of simplicity, in the following we will only discuss the special case where they are scalars. This choice does not result in any loss of generality.

**Definition B.1.** A partitioning of $D_T$, with the number of partitions denoted by $Z$, is a sequence of disjoint sets $A_1, A_2, \ldots$ so that their union yields $D_T$. That is: $\bigcup_{z=1}^{Z} A_z = D_T$.

A specific partition $A_z$ of $D_T$ is hence given by $A_z = \{(s_t, o_t) \mid t \in 1, 2, \ldots, T \text{ and } 1_{(o_t, s_t) \in A_z} = 1\}$, where $1_{(o_t, s_t) \in A_z}$ is
an indicator function that returns 1 if the criteria for the desired partition are met by the datapoint at $t$ and 0 otherwise. We



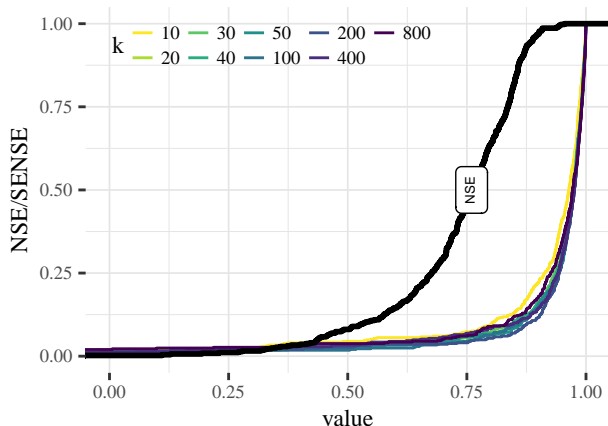

**Figure A1.** Approximation behavior of the SENSE, given the LSTM ensemble from Kratzert et al. (2019). The test data for a basin comprises 10 years of daily data (i.e., 3650 data points). Thus, 800 neighbors consider approximately 20% of the data for each time step. Note that the scale of the x-axis is chosen in a way that the NSE appears as a straight vertical line (since its values are very high in comparison).

will always assume that the partitions are chosen so that a given performance criterion can be reasonably evaluated. To give an example: for the NSE this would mean that no partition with less than two data-points can be created (since then the sample variance of the observations would be undefined), and it is not allowed to choose the partitions so that all observation values are the same (again, because then the sample variance of the observation would be zero). For any practical application this is
not a strong assumptions.

The indicator function for constructing $A_z$ trivially implies $A_z \subseteq D_T$ and leads to a convenience index function $\mathbb{I}_{A_z}$ that we will use to sum over the properties of a partition:

$$\mathbb{I}_{A_Z}(x) = \sum_{t=1}^{T} 1_{(o_t, s_t) \in A_z} * x_t, \tag{B1}$$

where $x$ here is a placeholder variable. We also define the model error as $e_t = o_t - s_t$, and will, for convenience, omit the
function arguments and brackets where it is clear from the context. For example:

1. $\mathbb{I}_{A_Z}(y) = \sum_{t=1}^{T} 1_{(o_t, s_t) \in A_z} * y_t$.

2. $\mathbb{I}_{A_Z}(1) = \sum_{t=1}^{T} 1_{(o_t, s_t) \in A_z}$ is the number of all elements in $A_z$ (that is, the size of the set $A_z$).

3. $\mathbb{I}_{A_Z} e_t := \mathbb{I}_{A_Z}(e_t) = \sum_{t=1}^{T} 1_{(o_t, s_t) \in A_z} * e_t$ is the sum of the errors (i.e., the bias) from the elements in $A_z$.





## B1 Generality

Without loss of generality we will prove all the below properties with **two** partitions for the sake of simplicity. The results generalize to higher numbers of partitions by recursively applying the same logic.

To illustrate this, let $L$ denote a performance criterion (where we use $L_{A_z}$ to express that we evaluate over the data in $A_z$), which obeys

$$L_{A_i} \leq L_{A_j} \implies L_{D_T} \leq L_{A_j}, \tag{B2}$$

where $A_i$ and $A_j$ are two partitions of the dataset $D_T$. In words, assumption B2 enforces that if the loss for one partition is smaller than the loss for a second one, then this implies that the loss for the overall data will also be smaller for the second partition.

**Proposition B.2.** *This inequality also holds if we divide $D_T$ into three partitions $A_a, A_b, A_c$. That is:*

$$L_{A_a} \leq L_{A_b} \leq L_{A_c} \implies L_{D_T} \leq L_{A_c} \tag{B3}$$

*also holds.*

*Proof.* To prove this, we use the fact that, by assumption B2, if $L_{A_a} \leq L_{A_b}$ then this implies that $L_{A_a \cup A_b} \leq L_{A_b}$. Formally,

$$L_{A_a} \leq L_{A_b} \implies L_{A_a \cup A_b} \leq L_{A_b},$$

and therefore

$$L_{A_a} \leq L_{A_b} \leq L_{A_c} \implies L_{A_a \cup A_b} \leq L_{A_b} \leq L_{A_c}.$$

Further,

$$L_{A_a \cup A_b} \leq L_{A_c} \implies L_{A_a \cup A_b \cup A_c} \leq L_{A_c}$$

also holds by assumption B2, so we can conclude that Eq. (B3) is true. □

This nested form of evaluation can be repeated no matter how many partitions there are and it can also be repeated for different implications. Therefore, we can analyze the behavior of two partitions without loss of generality.

## B2 Mean squared error

The sample MSE for a partition $A_X \in D_T$ is defined as:

$$\text{MSE}(A_x) = \frac{\mathbb{I}_{A_X} e_t^2}{\mathbb{I}_{A_X}}. \tag{B4}$$

**Proposition B.3.** *Given two partitions, $A_i \subset D_T$ and $A_j \subset D_T$, with $\text{MSE}_{A_i} \leq \text{MSE}_{A_j}$, the $\text{MSE}_{D_T}$ is bound by $\text{MSE}_{A_i} \leq \text{MSE}_{D_T} \leq \text{MSE}_{A_j}$.*



*Proof.* Expanding and rearranging $\text{MSE}_{A_i} \leq \text{MSE}_{A_j}$ gives:

$$\mathbb{I}_{A_J} * \mathbb{I}_{A_I} e_t^2 \leq \mathbb{I}_{A_I} * \mathbb{I}_{A_J} e_t^2.$$

If we expand on both sides by $\mathbb{I}_{A_I} * \mathbb{I}_{A_I} e_t^2$, we get:

$$\mathbb{I}_{A_J} * \mathbb{I}_{A_I} e_t^2 + \mathbb{I}_{A_I} * \mathbb{I}_{A_I} e_t^2 \leq \mathbb{I}_{A_I} * \mathbb{I}_{A_J} e_t^2 + \mathbb{I}_{A_I} * \mathbb{I}_{A_I} e_t^2,$$

$$(\mathbb{I}_{A_J} + \mathbb{I}_{A_I}) * \mathbb{I}_{A_I} e_t^2 \leq \mathbb{I}_{A_I} * \mathbb{I}_{D_T} e_t^2,$$


$$\frac{\mathbb{I}_{A_I} e_t^2}{\mathbb{I}_{A_I}} \leq \frac{\mathbb{I}_{D_T} e_t^2}{\mathbb{I}_{D_T}},$$

$$\text{MSE}_{A_i} \leq \text{MSE}_{D_T}.$$

Hence, the smaller MSE of the two partitions is also smaller than the MSE of the whole dataset. Inversely, the larger of the MSEs of the two partitions is larger than the MSE of the whole dataset — which can be shown analogously to the provided derivation. □

Thus, we can summarize the results of our proof with following relationship:

$$\text{MSE}_{A_i} \leq \text{MSE}_{D_T} \leq \text{MSE}_{A_j}.$$

## B3 Weighted mean squared error

The sample weighted mean squared error WMSE for a partition $A_X \in D_T$ is defined as:

$$\text{WMSE}(A_x) = \frac{\mathbb{I}_{A_X} w_t * e_t^2}{\mathbb{I}_{A_X} w_t}, \tag{B5}$$

where $w_t$ are the weights given to each individual sample.

*Proof.* The proof is analogous to proof B2. Despite the redundancy we show it in the following for the sake of completeness.

Expanding and rearranging $\text{WMSE}_{A_i} \leq \text{WMSE}_{A_j}$ gives:

$$\mathbb{I}_{A_J} w_t * \mathbb{I}_{A_I} w_t e_t^2 \leq \mathbb{I}_{A_I} w_t * \mathbb{I}_{A_J} w_t e_t^2.$$

If we expand on both sides by $\mathbb{I}_{A_I} w_t * \mathbb{I}_{A_I} w_t e_t^2$, we get:

$$\mathbb{I}_{A_J} w_t * \mathbb{I}_{A_I} w_t e_t^2 + \mathbb{I}_{A_I} w_t * \mathbb{I}_{A_I} w_t e_t^2 \leq \mathbb{I}_{A_I} w_t * \mathbb{I}_{A_J} w_t e_t^2 + \mathbb{I}_{A_I} w_t * \mathbb{I}_{A_I} w_t e_t^2,$$

$$(\mathbb{I}_{A_J} w_t + \mathbb{I}_{A_I} w_t) * \mathbb{I}_{A_I} w_t e_t^2 \leq \mathbb{I}_{A_I} w_t * \mathbb{I}_{D_T} w_t e_t^2,$$

$$\frac{\mathbb{I}_{A_I} w_t e_t^2}{\mathbb{I}_{A_I} w_t} \leq \frac{\mathbb{I}_{D_T} w_t e_t^2}{\mathbb{I}_{D_T} w_t},$$

$$\text{WMSE}_{A_i} \leq \text{WMSE}_{D_T}.$$

Hence, the smaller WMSE of the two partitions is also smaller than the WMSE of the whole dataset. Inversely, the larger of

the WMSEs of the two partitions is larger than the WMSE of the whole dataset — which can be shown analogously to the provided derivation. □





**Proposition B.4.** *Given two partitions, $A_i \subset D_T$ and $A_j \subset D_T$, with $\mathrm{WMSE}_{A_i} \leq \mathrm{WMSE}_{A_j}$, the $\mathrm{WMSE}_{D_T}$ is bound by $\mathrm{WMSE}_{A_i} \leq \mathrm{WMSE}_{D_T} \leq \mathrm{WMSE}_{A_j}$.*

### B4 Low Effort Nash-Sutcliffe Efficiency

For a partition $A_x \in D_T$ the LENSE is defined as:

$$\mathrm{LENSE}_{A_X} = 1 - \frac{\frac{1}{\mathbb{I}_{A_X}}\mathbb{I}_{A_X} e_t^2}{\frac{1}{\mathbb{I}_{X_R}}\mathbb{I}_{X_R} e_t^2}, \tag{B6}$$

where $X_R$ defines the data of the reference partition.

**Proposition B.5.** *Given two partitions, $A_i \subset D_T$ and $A_j \subset D_T$, with $\mathrm{LENSE}_{A_i} \leq \mathrm{LENSE}_{A_j}$, the $\mathrm{LENSE}_{D_T}$ is bound by $\mathrm{LENSE}_{A_i} \leq \mathrm{LENSE}_{D_T} \leq \mathrm{LENSE}_{A_j}$.*

*Proof.* We rewriting Eq. (B6) in terms of MSE as

$$\mathrm{LENSE}_{A_X} = 1 - \frac{\mathrm{MSE}_{A_X}}{\frac{1}{\mathbb{I}_{X_R}}\mathbb{I}_{X_R} e_t^2}. \tag{B7}$$

By inserting Eq. (B7) into the inequality from proposition B.5 and rearranging we get:

$$\frac{\mathrm{MSE}_{A_i}}{\frac{1}{\mathbb{I}_{X_R}}\mathbb{I}_{X_R} e_t^2} \geq \frac{\mathrm{MSE}_{D_T}}{\frac{1}{\mathbb{I}_{X_R}}\mathbb{I}_{X_R} e_t^2} \geq \frac{\mathrm{MSE}_{A_j}}{\frac{1}{\mathbb{I}_{X_R}}\mathbb{I}_{X_R} e_t^2}. \tag{B8}$$

Here, the denominator is a constant (we can also relate it to the WMSE by observering that $w_t = w = \frac{1}{\frac{1}{\mathbb{I}_{X_R}}\mathbb{I}_{X_R} e_t^2}$). Thus, we can reduce the equation to obtain:

$$\mathrm{MSE}_{A_i} \geq \mathrm{MSE}_{D_T} \geq \mathrm{MSE}_{A_j}, \tag{B9}$$

which we know to be true from proof B2. □

### B5 Situation-equitable Nash–Sutcliffe Efficiency

The SENSE for a partition $A_x \in D_T$ is defined as:

$$\mathrm{SENSE}_{A_X} = 1 - \frac{\mathbb{I}_{A_X} \frac{1}{(o_t - \hat{\mu}_t)^2}(o_t - s_t)^2}{\mathbb{I}_{A_X} \frac{1}{(o_t - \hat{\mu}_t)^2}}, \tag{B10}$$

where $\hat{\mu}_t$ is an estimation for the conditional expectation of the observations of at timestep $t$. Appendix A discusses a potential approach for obtaining such an estimation.

**Proposition B.6.** *Given two partitions, $A_i \subset D_T$ and $A_j \subset D_T$, with $\mathrm{SENSE}_{A_i} \leq \mathrm{SENSE}_{A_j}$, the $\mathrm{SENSE}_{D_T}$ is bound by $\mathrm{SENSE}_{A_i} \leq \mathrm{SENSE}_{D_T} \leq \mathrm{SENSE}_{A_j}$.*

*Proof.* It is easy to see that the right hand side of Eq. (B10) is a special case of the WMSE with $w = \frac{1}{}$. Thus, the proof is analogous to proof B3. □





## B6 Nash–Sutcliffe Efficiency

The NSE for a partition $A_X \in D_T$ is defined as:

$$\mathrm{NSE}_{A_X} = \frac{\mathbb{I}_{A_X} e_t^2}{\bar{o}_X}. \tag{B11}$$

**Proposition B.7.** *Given two partitions, $A_i \subset D_T$ and $A_j \subset D_T$, with $\mathrm{NSE}_{A_j} \geq \mathrm{NSE}_{A_i}$; $\mathrm{NSE}_{D_T}$ is bound by below by the smaller* NSE *value of the two partitions — i.e., $\mathrm{NSE}_{A_i}$.*

*Proof.* Following the convention from B.3 and Eq. (B11), we get:

$$1 - \frac{\mathbb{I}_{A_J} e_t^2}{\mathbb{I}_{A_J}(o_t - \bar{o}_J)^2} \geq 1 - \frac{\mathbb{I}_{A_I} e_t^2}{\mathbb{I}_{A_I}(o_t - \bar{o}_J)^2}.$$

By subtracting 1, multiplying by $-1$, and rearranging $\mathbb{I}_{A_J} e_t^2$ and $\mathbb{I}_{A_I} e_t^2$ we get:

$$\frac{\mathbb{I}_{A_J} e_t^2}{\mathbb{I}_{A_I} e_T^2} \leq \frac{\mathbb{I}_{A_J}(o_t - \bar{o}_J)^2}{\mathbb{I}_{A_I}(o_t - \bar{o}_I)^2},$$

adding 1 to both sides and substituting $e_t$ gives:

$$\frac{\mathbb{I}_{A_J} e_t^2}{\mathbb{I}_{A_I} e_t^2} + 1 \leq \frac{\mathbb{I}_{A_J}(o_t - \bar{o}_J)^2}{\mathbb{I}_{A_I}(o_t - \bar{o}_I)^2} + 1,$$

and

$$\frac{\mathbb{I}_{A_J} e_t^2}{\mathbb{I}_{A_I} e_t^2} + \frac{\mathbb{I}_{A_I} e_t^2}{\mathbb{I}_{A_I} e_t^2} \leq \frac{\mathbb{I}_{A_J}(o_t - \bar{o}_J)^2}{\mathbb{I}_{A_I}(o_t - \bar{o}_I)^2} + \frac{\mathbb{I}_{A_J}(o_t - \bar{o}_J)^2}{\mathbb{I}_{A_J}(o_t - \bar{o}_J)^2}. \tag{B12}$$

At this stage we note that $\bar{o}(A_i)$ minimizes the squared distance for the samples in $A_i$, and vice versa $\bar{o}(A_j)$ minimizes for samples in $A_j$. Any other choice than $\bar{o}(A_i)$ (or $\bar{o}(A_j)$, respectively) in their respective partitions leads to a larger sum. For example:

$$\mathbb{I}_{A_J}[o_t - \bar{o}_J)]^2 + \mathbb{I}_{A_I}[o_t - \bar{o}_I)]^2 \leq \mathbb{I}_{A_J}[o_t - \bar{o}_T]^2 + \mathbb{I}_{A_I}[o_t - \bar{o}_T]^2,$$

which we can use to bind the right-hand side of Eq. (B12). After some rearrangement, we have

$$\frac{\mathbb{I}_{D_T} e^2}{\mathbb{I}_{A_I} e^2} \leq \frac{\mathbb{I}_{D_T}[o_t - \bar{o}_T]^2}{\mathbb{I}_{A_I}[o_t - \bar{o}_I)]^2}.$$

From here we can manipulate the equation to reintroduce the canonical formulation of the NSE and we get:

$$1 - \frac{\mathbb{I}_{D_T} e_t^2}{\mathbb{I}_{D_T}(o_t - \bar{o}_T)^2} \geq 1 - \frac{\mathbb{I}_{A_I} e_t^2}{\mathbb{I}_{A_I}(o_t - \bar{o}_I)^2},$$

$$\mathrm{NSE}_{D_T} \geq \mathrm{NSE}_{A_i}.$$

Thus, the lower NSE of the two partitions is the lower bound of the NSE of the whole dataset. □

This means that that we obtain the following relation for the NSE:

$$\mathrm{NSE}_{A_i} \leq \mathrm{NSE}_{D_T} \leq 1.$$





## B7 Pearson's correlation coefficient

The sample correlation coefficient over a partition $A_X \subseteq D_T$ is

$$r_{A_X} = \frac{\mathbb{I}_{A_X}(o_t - \bar{o}_X)(s_t - \bar{s}_X)}{\sqrt{\mathbb{I}_{A_X}(o_t - \bar{o}_X)^2}\sqrt{\mathbb{I}_{A_X}(s_t - \bar{s}_X)^2}}. \tag{B13}$$

470 In most cases, $r_{A_z}$ is independent from $r(D_T)$ — as is demonstrated by Simpson's Paradox (Sec. 1) — and we can only make claims for special cases. For example, if $r(D_T) = 1$ (or $-1$) then it follows that $r_{A_z} = 1$ (or $-1$). Similarly, the following proposition shows that if we assume the partitions have zero mean for the observations ($\bar{o}_X = 0$) and the simulations ($\bar{s}_X = 0$), then $r_{A_z}$ is lower-bounded by $r(D_T)$:

**Proposition B.8.** *If we assume that all partitions have centered observations and model outputs, we can bind the sample*
475 *correlation from above by the higher one; i.e.: $r(D_T) \leq r_{A_j}$.*

*Proof.* By applying the assumption from proposition B.8 to Eq. (B13), we get:

$$r_{A_z} = \frac{\mathbb{I}_{A_X}(o_t)(s_t)}{\sqrt{\mathbb{I}_{A_X}(o_t)^2}\sqrt{\mathbb{I}_{A_X}(s_t)^2}}.$$

Using $r_{A_i} \leq r_{A_j}$ results in:

$$\frac{\mathbb{I}_{A_I}(o_t)(s_t)}{\sqrt{\mathbb{I}_{A_I}(o_t)^2}\sqrt{\mathbb{I}_{A_I}(s_t)^2}} \leq \frac{\mathbb{I}_{A_J}(o_t)(s_t)}{\sqrt{\mathbb{I}_{A_J}(o_t)^2}\sqrt{\mathbb{I}_{A_J}(s_t)^2}},$$

$$\frac{\mathbb{I}_{A_I}(o_t)(s_t)}{\mathbb{I}_{A_J}(o_t)(s_t)} \leq \frac{\sqrt{\mathbb{I}_{A_I}(o_t)^2}\sqrt{\mathbb{I}_{A_I}(s_t)^2}}{\sqrt{\mathbb{I}_{A_J}(o_t)^2}\sqrt{\mathbb{I}_{A_J}(s_t)^2}},$$

$$\frac{\mathbb{I}_{A_I}(o_t)(s_t) + \mathbb{I}_{A_J}(o_t)(s_t)}{\mathbb{I}_{A_J}(o_t)(s_t)} \leq \frac{\sqrt{\mathbb{I}_{A_I}(o_t)^2}\sqrt{\mathbb{I}_{A_I}(s_t)^2} + \sqrt{\mathbb{I}_{A_J}(o_t)^2}\sqrt{\mathbb{I}_{A_J}(s_t)^2}}{\sqrt{\mathbb{I}_{A_J}(o_t)^2}\sqrt{\mathbb{I}_{A_J}(s_t)^2}},$$

$$\frac{\mathbb{I}_{D_T}(o_t)(s_t)}{\mathbb{I}_{A_J}(o_t)(s_t)}, \leq \frac{\sqrt{\mathbb{I}_{D_T}(o_t)^2}\sqrt{\mathbb{I}_{D_T}(s_t)^2}}{\sqrt{\mathbb{I}_{A_J}(o_t)^2}\sqrt{\mathbb{I}_{A_J}(s_t)^2}},$$

hence:

485 $r_{D_T} \leq r_{A_j}.$

Thus, if all data (observations and model output) in the partitions is centered, then the higher correlation of the two partitions is an upper bound for the overall correlation. □

## Appendix C: More experimental results

### C1 Model evaluation with the NSE and the LENSE for different models

490 This section shows alterations of the first part of our analysis (Sect. 2.2) using different models.





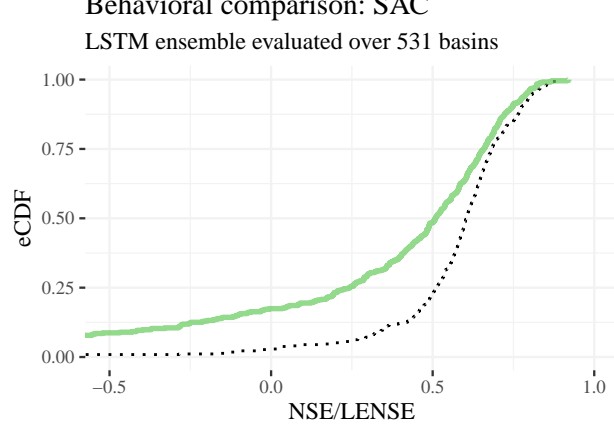

**Figure C1.** Empirical cumulative distribution functions of the NSE (black, dotted line) and the LENSE (green line) for the 531 CAMELS basins and an ensemble of calibrated HBV models from (see Kratzert et al., 2019).

**Figure C2.** Empirical cumulative distribution functions of the NSE (black, dotted line) and the LENSE (green line) for the 531 CAMELS basins and the SAC-SMA model (see Kratzert et al., 2019).





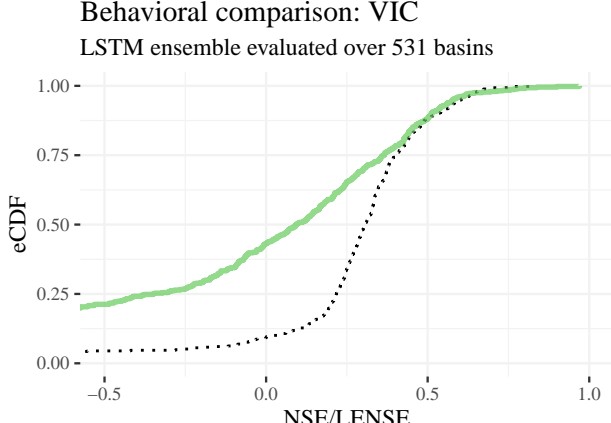

**Figure C3.** Empirical cumulative distribution functions of the NSE (black, dotted line) and the LENSE (green line) for the 531 CAMELS basins and the VIC-conus model (see Kratzert et al., 2019).

## C2 Two more experimental results

This appendix shows two additional sweeps of our experiment from Sec. 2.1. One has $NSE_{low} = 0.25$ (Fig. C4) and the other $NSE_{low} = 0.75$ (Fig. C5).

*Author contributions.* DK had the initial idea for the paper. DK and MG set up the experiment (and also many negative results along the way). MG came up with the first version of the SENSE criterion. DK and MG realized the theoretical appendix. DK, GN, and JZ conceptualized the paper structure. All authors contributed to the analysis of the results, the discussion of the interpretations, and the creation of the figures, and the writing process,

## Appendix D: Flow–duration curve based metrics and the DAMN.

This appendix provides a short comment on why many of the currently used metrics based on flow–duration curves do not guard against the DAMN. Specifically, we discuss the case of the percent bias of the bottom 30% low flow range (FLV) and percent bias of the top 2% high flow range (FHV) as defined in Yilmaz et al. (2008). Both the FLV and the FHV first divide the data based on the flow-duration curve and then compute the percent bias for the flow–duration values that fall within the predefined partition. This approach has three problems with regard to the DAMN:

1. The a-priori set thresholds (30% and 2%) are too coarse to capture situational differences in model performance. For example, a model that captures rain–driven high-flows well, but melt–driven ones badly might still exhibit a good FHV if the former occur frequently enough to fall over the threshold. This problem is made worse by point 2.





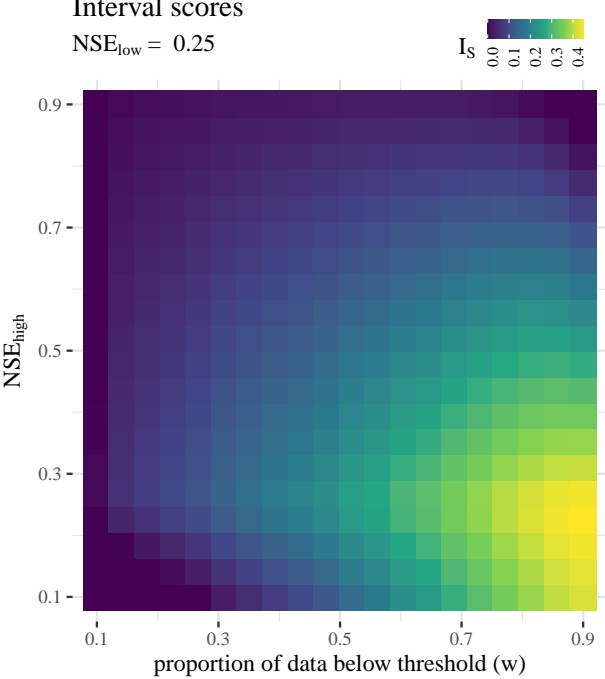

**Figure C4.** Rerun of our experiment (Sec. 2.1) with $NSE_{low}$ fixed at 0.25. Each pixel in the plot represents an "interval score", which is zero if the overall model performance $NSE_{all}$ is within the interval spanned by model performance in the low-flow partition, $NSE_{low}$, and the high-flow partition, $NSE_{high}$. In the other case, the interval score is negative if $NSE_{all}$ is lower than the interval, and positive if $NSE_{all}$ is higher (see Fig. 3).

2. The relative bias can be compensated by varying situational performance. For example, a model that overestimates some set of peaks, but equally underestimates another set of peaks can have an FHV that is close to 0 (nearly perfect) — despite the model performing badly for all peaks. This problem is exacerbated by point 3.

510  3. The flow duration curve breaks temporal locality: The temporal occurrence of the events is not considered, so the behavior in one situation can compensate for the behavior in another. For example, if a model underestimates the highest peak in the data but overestimates another unrelated event, the FHV can be close to zero (i.e., nearly perfect).

*Competing interests.* The authors declare no competing interests.

*Acknowledgements.* We are grateful for the support and guidance of Sepp Hochreiter, who is always generous with his time and ideas.
515  We would also thank Lukas Gruber for insightful discussions regarding our theoretical considerations of DAMN. Due to his critical eye





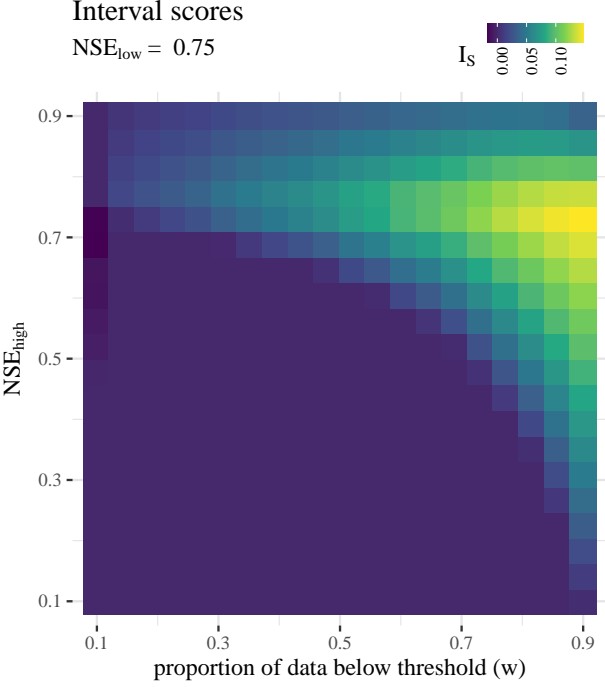

**Figure C5.** Rerun of our experiment (Sec. 2.1) with $NSE_{low}$ fixed at 0.75. Each pixel in the plot represents an "interval score", which is zero if the overall model performance $NSE_{all}$ is within the interval spanned by model performance in the low-flow partition, $NSE_{low}$, and the high-flow partition, $NSE_{high}$. In the other case, the interval score is negative if $NSE_{all}$ is lower than the interval, and positive if $NSE_{all}$ is higher (see Fig. 3).

the Appendix became much more thorough. Further, we need to mention Claus Hofman, Andreas Radler, and Annine Duclaire Kenne for bouncing off ideas for the experiments, even if they ultimately did not materialize as we imagined.