# Peer review of "Technical Note: The divide and measure nonconformity — How metrics can mislead when we evaluate on different data partitions."

_Hydrology and Earth System Sciences, 2024_

## Referee Comment (RC2)

**Technical Note: The Divide and Measure Nonconformity**

Daniel Klotz[1], Martin Gauch[2], Frederik Kratzert[3], Grey Nearing[4], and Jakob Zscheischler[1,5]

[1]Department of Compound Environmental Risks, Helmholtz Centre for Environmental Research — UFZ, Leipzig, Germany
[2]Google Research, Zurich, Switzerland
[3]Google Research, Vienna, Austria
[4]Google Research, Mountain View, California, USA
[5]Technische Universität Dresden, Dresden, Germany

**Correspondence:** Daniel Klotz (daniel.klotz@ufz.de)

**Abstract.** The evaluation of model performance is an essential part of hydrological modeling. However, leveraging the full information that performance criteria provide, requires a deep understanding of their properties. This Technical Note focuses on a rather counterintuitive aspect of the perhaps most widely used hydrological metric, the Nash-Sutcliffe Efficiency (NSE). Specifically, we demonstrate that the overall NSE of a dataset is not bounded by the NSEs of all its partitions. We term

5    this phenomenon the "Divide and Measure Nonconformity". It follows naturally from the definition of the NSE, yet because modelers often subdivide datasets in a non-random way, the resulting behavior can have unintended consequences in practice. In this note we therefore discuss the implications of the "Divide and Measure Nonconformity", examine its empirical and theoretical properties, and provide recommendations for modelers to avoid drawing misleading conclusions.

**1 Introduction**

10    Measuring model performance is a foundational pillar of environmental modeling. For instance, in order to assure that a model is suited rainfall-runoff, we have to test how "good" it is. Over time, our community has established a set of performance criteria that cover different aspects of modelling. These criteria allow us to draw conclusions with regard to the evaluated model and should therefore exhibit consistent behaviour that follows our intuitions as modelers. However, when we use these criteria it is important to keep in mind that each one has specific properties — certain advantages and disadvantages — that are

15    relevant for interpreting results.

The Nash–Sutcliffe Efficiency (NSE; Nash and Sutcliffe, 1970) is the perhaps most used metrics in hydrology. In this contribution we show that the NSE exhibits a counterintuitive behavior (which, as far as we can tell, is so far undocumented), captured by the following exemplary anecdote. A hydrologist evaluates a model over a limited period of time and obtains an NSE value of, say, 0.77 (Fig. 1, blue partition). Then, a large event occurs and an isolated evaluation for that specific event

20    results in the slightly worse model performance of, say, 0.75 (Fig. 1, orange partition). One would then expect that the overall performance (i.e., a model evaluation over both the the blue and the orange partitions) should be bound by the values obtained during evaluation over each partition separately. However, the NSE over the entire time series in this example is 0.80 (Fig. 1, purple partition), which is higher than either partition.

[Figure]

**Figure 1.** Example of the part–whole relationship within the Divide and Measure Nonconformity. The blue data partition has an NSE of 0.77, the orange data partition (that contains the peak event) has an NSE of 0.75. However, he overall NSE is 0.8 (violet partition), which is larger than both individual partitions.

We refer to the phenomenon that the overall NSE can be higher than the NSEs of data subdivisions as the *Divide and Measure Nonconformity* (DAMN). A natural question that follows from here is: What is the cause for the "DAMN behavior" in the example? To give an answer it is useful to consider the formal definition of the NSE:

$$\text{NSE} = 1 - \frac{\sum_{t=1}^{T}(o_t - s_t)^2}{\sum_{t=1}^{T}(o_t - \bar{o}_t)^2}, \qquad \text{(1)}$$

bar should be over O_t since (supposedly) Obar does not vary with time, or else use Obar_T

[revised manuscript text omitted]
 lower values because under these conditions the (associated) mean is a good estimator, and therefore indicates a "benchmark" that is harder to improve upon.*

**Generality.** In this Technical Note we restrict our discussion to the NSE because of its importance and ubiquity. However, the *phenomenon* DAMN is a general phenomena and can occur for other performance criteria. For example, the Kling-Gupta Efficiency (KGE; Gupta et al., 2009) exhibits similar empirical behavior (we do not show this explicitly in this note, but encourage readers to explore it, e.g., by using our code, which provides an implementation of the KGE for testing). That said, simple average-based metrics such as MSE are not subject to the DAMN (see Appendix B2). *Because they are not "benchmarked" in the same manner …*

*NOTE: This suggests that the problem will persist whenever one uses a locally-sourced benchmark to compute the normalized metric. Would a way around this problem be to always standardize the data in each partition to have the same mean and/or range (I did not think this through)? More generally, if we "understand" that the locally-benchmarked metric indicates only locally-relevant indication of normalized performance, then in principle there is no problem … :-? Perhaps this point could be discussed / made more clear. The problem arises mainly when trying to compare across data splits.*

[revised manuscript text omitted]

---

## Referee Comment (RC3)

[referee-annotated manuscript omitted]

---

## Author Comment (AC1)

Example depiction for the adaption of Fig. 2 in the revised manuscript:

[Figure]

[Figure]

Example depiction for the adaption of Fig. 6 in the revised manuscript:

---

## Author Response (AR1)

**DAMN review answers**

**Reviewer 1**

We thank the reviewer for the thorough review and constructive feedback. In the following, we provide our answers to the individual comments (where the original comments are written bold).

Klotz et al. elucidate the Divide and Measure Nonconformity (aka DAMN) behavior of the NSE-metric. The DAMN can be summarized as; the NSE of a data-series is always higher or equal to the NSE of any subset of this data-series (i.e., the overall NSE of a dataset is not bounded by the NSEs of all its partitions). This behavior not only occurs for the NSE, but for any performance measure that is standardized by observed variance.

Indeed, I was not aware of this behavior, and I deem it relevant for hydrological modellers to be aware of it, and as such I think the very existence of this Technical Note is well justified. The authors provide proof for the DAMN behavior and describe the implications of this behavior.

Although I have little feedback on the technical core of this paper, I think the discussion and presentation can be improved for readability and to increase the relevance. Below, I provide some suggestions that might help the authors.

Firstly, it would be more natural to me to move some of the "related work", namely the whole section on "data splitting", to a-newly-to-be-added Discussion. The same applies to all the sections under "1.2 Relevance" (besides it seems that "Settings" treats comparable topics as "Data splitting"). The Discussion could then be further enhanced by providing concrete implications and recommendations. Should we no longer use the differential split sample? Are the different NSEs inherently incomparable (I don't think so – if one doesn't want to compare model performance to mean observations one should use (R)MSE), or might it just be misleading if you're not aware of this behavior?).

A similar idea was also raised by reviewer Wouter Knoben, and we do indeed agree that a separated discussion section would be very beneficial for the manuscript. Therefore, we will restructure the manuscript in the proposed way and include a discussion on the recommendations. Our goal for this extended discussion will be the same as for the rest of the paper: To bring more nuance into model evaluation exercises.  To shortly answer the reviewer questions here specifically: We believe that differential split sampling should remain a part of the hydrological mode model building toolbox, however modelers should be aware of the DAMN when evaluating their models on differential splits. Similarly, we argue that different NSE values are often somewhat comparable. The DAMN phenomenon makes it so that comparisons get more difficult when the samples are split into partitions with widely different statistical properties (say, rivers/periods with very low variance and rivers/periods with very high variance in streamflow). However, from the "DAMN perspective" it is even then not necessarily wrong to compare the NSEs, modelers should just be aware that these will then not tell the whole story.

On the same page, I was wondering if the DAMN phenomenon could be linked to calibration studies. Many studies evaluate the required period for calibration, and conclude that a longer calibration period generally leads to better results (at least for the calibration-period, of course when that is compared to the validation period the DAMN does not apply anymore). This, then, is a combination of the information in the time series that enhances the calibration, and the effect of the DAMN-phenomenon. The relative effect of these two can be disentangled with the LENSE-tool presented here. I was also wondering if the DAMN-phenomenon could have implications for conclusions on overparameterization, but probably not since for this, mostly two independent time periods are used.

This is an interesting observation. Indeed, when comparing model performance with increasing calibration and validation period lengths, the performance changes will be due to a mix of changes in the actual model quality and of DAMN effects. Therefore, we believe that such evaluations should only increase the *calibration* period length but keep a fixed validation period. We are not sure which specific relations to overparameterization the reviewer has in mind, but we also agree that

calibration and validation periods are typically (and should always be) entirely separate.

From the methods in 2.1 it is not entirely clear why NSElow is fixed to a prescribed value while NSEhigh can vary. After reading the results, I think it does not matter how one would approach it (either fix the high or the low) but this remains a bit unclear in het methods.

This is correct. Our analysis does not depend on which of the two NSEs is fixed. We chose to fix the lower one, since we were thinking about the risk that a model performs well on most flows but provides bad predictions for peak flows (in our experience this happens easily and the disentangling of performance for specific situations is not simple, as the DAMN shows). We did, however, never describe this explicitly in the manuscript and will add an explanation of the arbitrariness of the choice and a short motivation for it.

I know this is a bit generic, but I have the feeling that 3.1 is not so clearly worked out (hard to follow / to be convinced). Maybe it would help to include some of the arguments treated in Appendix B.

We would like to keep the theoretical treatment in the appendix, but we also do not want readers to get the feeling that a section of the paper is not well worked out. We therefore propose that we extend the section with a discussion of the wider implications of it and not just focus on the specific results of the experiments.

Concerning presentation / figures:
Figure 2: I suppose there is a little hint in showing that surface hydrology class needs fewest preparation hours for the highest grade.. To further emphasize the point of this graph you could add the linear regression line to both panels (1 in the upper and 3 in the lower one).

We will add regression lines and change the joke/hint since it was not meant to be offensive (we will exchange it with a less risky alternative that applies to many areas of research; see attached image) .

[Figure]

Figure 6 is unclear: what is the yellow line and what are the grey lines? Besides there is a typo in the figure-title (evalauted>evaluated). More in general I would suggest to remove the figure titles, this information can be added/is already present in the caption .

We will adapt the proposed changes. The color of the lines reflect the difference between NSE and LENSE and go from violet to yellow. The gray lines indicate an explosion of the NSE towards negative infinity, which we clipped at -0.5. However, as the comment rightly points out we did indeed forget to describe any of this  (a similar concern was raised in a minor comment from reviewer Wouter). For the revised version we will therefore simplify the plot by truncating first, add a legend to the image and improve the figure caption (see attached document for an example).

[Figure]

Textual / minor

l. 11 is suited rainfall-runoff -> is suited to simulate rainfall-runoff behavior

Thank you.  We will change this part to "is suited to simulate the rainfall–runoff relationship" just because we use "behavior" quite often later in a different context.

l. 10 -15 Note that the criteria you refer to here are meant to evaluate model output, not so much the model itself (because it might depend a lot on implementation and also on data issues). Therefore perhaps good to reword slightly to be more exact on this.

Good point. We will rephrase this slightly.

We will change this.

:>

We will do so.

We will change this.

We agree that the sentence is a bit involved, but we feel it is important since we want to already give readers a feeling for the logic behind the LENSE in this part of the manuscript. We will rephrase the sentences in the following way to make it clearer:

One can imagine taking the squared error term (the numerator of the NSE metric) over only the second (orange) partition, but using the observational variance (the denominator of the NSE metric) from the whole (purple) time period. This would result in a value higher than the actual NSE in the second (orange) period.

Yes indeed.

We will change this.

l. 245-246 Interesting suggestion but the implications or what we learn from it are not directly clear

Thank you. It would make the behavior of the NSE (or whichever metric) clearer and possibly allow (quantitative) comparisons of the "flexibility" of different metrics. We will emphasize this in the revised version.

l. 325 remove one "that"

We will change this.

l. 343 and 349 for clarity perhaps good to keep the order of St and Ot consistent

We will change this.

Appendix A - It is a bit confusing that the mean-symbol is used to express a variance, or did I misunderstand something?

We will use sigma instead.

Figure A1 Caption: I don't understand the last sentence.

You are right, this is a leftover from an earlier version of the figure. We will remove the sentence.

Under B1 Generality, for completeness it would be good to clarify if the performance criterion needs to be maximized or minimized.

We will explicitly state that we assume that the performance criterion is minimized (and that this, again, does not result in a loss of generality).

l. 376 Where is Assumption B2?

Assumption B2 is shown in the form of equation (B2) (l. 369). We agree that this is confusing, because we do not delineate this assumption separately, but make it part of a proposition. In the revised manuscript we will refer to B2 as "the assumption from Eq. (B2)" and adapt l.370, l.376 and l.382 accordingly.

Author contributions seems to be placed incorrectly (either before or after all appendices)

We will move everything after the appendices.

l. 510 in the third argument I don't understand the last sentence.

If the model is good for event A but bad for event B the errors can compensate each other. We will rephrase this.

**Reviewer 2: Hoshin Gupta**

We thank the reviewer for the enthusiastic review! In the following, we will answer the major comments from the review. We agree with all the minor suggestions (not quoted below) and will adapt them as proposed by the reviewer.

[L. 92ff] This suggests that the problem will persist whenever one uses a locally-sourced benchmark to compute the normalized metric. Would a way around this problem be to always standardize the data in each partition to have the same mean and/or range (I did not think this through)? More generally, if we "understand" that the locally-benchmarked metric indicates only locally-relevant indication of normalized performance, then in principle there is no problem [...]? Perhaps this point could be discussed / made more clear. The problem arises mainly when trying to compare across data splits.

We thought about this interesting proposal and its implications and found that it will lead to an interesting special case, which we will cover in a dedicated subsection of the appendix (see attached document for an exemplary draft) replacing the incomplete discussion of the correlation coefficient. In short, it does seem that the proposed setting will not get rid of the DAMN per se, but focus the evaluation on comparing the rotational similarity of a vector of observations with a vector of simulations.

The attached document:

**B8 The special case of standardized data**

460 During the review process, reviewer Hoshin Gupta inspired us to think about what would happen to the NSE if the available data would always be standardized (i.e., both the observations and simulations have zero mean and unit variance for all partitions and the overall data). This section shows that in this special setting the NSE and the Kling-Gupta Efficiency (KGE just measures the Pearson's correlation coefficient $r$, and the correlation becomes the same as the cosine similarity.

**Proposition B.8.** *In a setting where we standardize the observations and model outputs for a given set of observations and* 465 *simulations, we get* $\mathrm{NSE} = 2 * r - 1$.

*Proof.* As per Gupta et al. (2009) the NSE can be decomposed into

$$\mathrm{NSE} = 2 * \alpha * r - \alpha^2 - \beta, \tag{B14}$$

where $\alpha$ is the ratio of the standard deviations, i.e.: $\alpha = \frac{\sigma_s}{\sigma_o}$, $r$ is Pearson's correlation coefficient, and $\beta = \frac{\mu_s - \mu_o}{\sigma_o}$ (here independent of the partition).

470 Since the means of the observations and simulations are zero, it always holds that $\beta = 0$ and $\alpha = 1$, which simplifies Eq. (B14) to

$$\mathrm{NSE} = 2 * r - 1.$$

In other words, in this special setting the NSE only measures the correlation. $\square$

There exists a similar simplification for the KGE:

475 **Proposition B.9.** *In a setting where we standardize the observations and model outputs, it holds that* $\mathrm{KGE} = r$.

*Proof.* The KGE is defined as

$$\mathrm{KGE} = 1 - \sqrt{(r-1)^2 + (\sigma_s/\sigma_o - 1)^2 + (\mu_s/\mu_o - 1)^2}.$$

In the current setting $\frac{\mu_s}{\mu_o}$ is actually undefined because of the division by zero, but we might also interpret it as one because $\mu_s = \mu_o$. Similarly, $\frac{\sigma_s}{\sigma_o} = 1$. Thus, the only part within the square root that remains is $(r-1)^2$, which gives us:

480
$$\begin{aligned}
\mathrm{KGE} &= 1 - \sqrt{(r-1)^2}, \\
&= 1 - |r-1|, \\
&= r.
\end{aligned}$$

Thus, we showed that in the special setting where observations and simulations are standardized the KGE measures the correlation only. $\square$

485 Next, we show that within the standardization setting the correlation becomes the cosine similarity.

**Proposition B.10.** *In a setting where we standardize the observations and model outputs for all data and all partitions, the correlation is the same as the cosine similarity.*

*Proof.* The cosine similarity between two $N$ dimensional vectors $a$ and $b$ is defined as

$$s_c = \cos\theta = \frac{\sum_{i=1}^{N} a_i b_i}{\sqrt{\sum_{i=1}^{N} a_i^2} * \sqrt{\sum_{i=1}^{N} b_i^2}}, \tag{B15}$$

490 where $\theta$ is the angle between the two vectors, and equivalence to $r$ is given because Eq. (B15) is the same as Eq. (B13) if the means are set to zero. $\square$

In other words, in this special setting the correlation just measures the difference in rotation given by the two centered vectors that contain the observations and the simulations respectively.

[L. 230ff] In Schaefli and Gupta (2007) we propose the need to use an "appropriate" "benchmark" for metric normalization (although not the excellent insights you provide here). Perhaps it might be worth mentioning that the "appropriate" benchmark can vary with year [...]. This might be interesting to explore, although it complicates matters by forcing us to think about what the appropriate benchmark prediction is ...

This is indeed important. The direction is similar to what we tried to explore with the idea of the SENSE criterion in that the reference model is always derived from the most similar situations (as expressed by the nearest neighbors of a runoff timeframe). The revised manuscript will mention this idea as potential future work.

**Reviewer 2: Wouter Knoben**

Thank you for the thorough review! In the following, we will answer the major comments from the review (quoted in bold). We agree with all the minor suggestions (not quoted below) and will adapt them as proposed by the reviewer.

*[Title] It may be worth considering an addition to this title to briefly outline what the divide and measure nonconformity is.*

We will expand the title to: The divide and measure nonconformity: how metrics can mislead when we evaluate on different data partitions.

*[L. 46] A key paper missing here is Lamontagne et al. (2020; [...]).*
*These authors draw a distinction between the true value E and its sample estimate NSE, and document NSE's very large variability from one sample to the next. See e.g. their Fig. 4 and accompanying text, as well as their Section 5.4. Their point that values such as NSE and KGE are sample estimates and thus inherently conditional on how the sample is selected, is closely related to your section on Data Splitting. I think it may also be worthwhile to point out Lamontagne's recommendation that NSE should be avoided for use with (sub-)daily data for the these reasons.*
*I may be wrong, but I believe the behaviour you describe in this paper may be a specific consequence of Lamontagne's more general findings about the behaviour of NSE.*
*I do not think that this invalidates the point of this Technical Note, because a more practical and more easily accessible discussion of these concepts is needed.*

The paper of Lamontagne et al. (2020) is indeed very relevant and we will include it in the related work. In our eyes it connected to our work in the same fashion as Clark et al. (2021). As such, we view this line of research as complementary to ours: The Lamontagne et al. (2020) line of work deals with problems of estimating a statistical property from small samples, while we explore the implications that partitioning or concatenation of data has for model evaluation. DAMN is not intrinsically a small sample problem, however the two problems can compound. That is, the DAMN phenomenon can appear or increase in small data regimes more easily because the errors in estimation will generally be larger (than in large or infinite data regimes).

[L.55f] I'm reasonably certain that this is the main point of an earlier paper (Schaefli & Gupta, 2007; https://onlinelibrary.wiley.com/doi/10.1002/hyp.6825) and is also implicitly mentioned in Seibert, 2001 (https://onlinelibrary.wiley.com/doi/epdf/10.1002/hyp.446)
We agree and will reference both of these papers.

[L.83] On my second read of the paper I think this section might be better placed as a general discussion section after the experiments and before the conclusions. I think the reader will have a better understanding of the phenomenon at that point and will be more easily able to follow the examples discussed here.
We agree and will adapt the manuscript accordingly.

[L.84] If my understanding of the problem is correct, I believe this generalizes to any study that compares DAMN-susceptible metrics over different periods and draws conclusions based on performance differences.
Yes, this is correct. In the revised version of the manuscript we will make the introduction statement stronger.

[L.163] Because we're looking at a sample estimate of variance, shouldn't the fraction $1/Tr$ in Eq.3 theoretically be $1/(Tr-1)$? This also applies to the NSE (e.g., Lamontagne et al., 2020, Eq. 5b), though I suppose in the NSE case it's slightly easier to assume that $(1/n)$ and $(1/(n-1))$ approximately cancel out for large enough n. For shorter reference periods Tr this might matter, though I find it hard to judge how much.
We understand where this idea is coming from, but have decided to stick to our original formulation. We would like to bring forth two reasons for our standpoint. For one, the proposed change would somewhat deflect from the purpose of the LENSE (Eq. 3) as a way of thinking about the model performance over an evaluation set with a performance reference model over a reference set. The second point is that the proposed correction would not make the (hypothetical) estimation unbiased either:

For our argument regarding the statistical appropriateness of the proposed change, let us adopt the formalism of Lamontagne et al. (2020) and interpret the LENSE as an approximation to a theoretical construct, which is the ratio of a expectation for a hypothetical distribution of squared errors and the variance of a hypothetical distribution of observations. In that case, it would generally *not* suffice to use unbiased estimators for the numerator and the denominator to get an unbiased estimator of the ratio!

To see this, consider a scenario where the numerator n and the denominator d are both 1, but we can only estimate d with an unbiased precision of ±0.5. In this case, if the estimation of the variance from given data would result in an underestimation of e, say, of ê=0.5, this will result in an overestimation of the actual ratio, since we would estimate it to be 2. However, an overestimation with the same delta, i.e., ê=1.5 will result in an estimated ratio of 0.66 (because we are only linear in log-space). Hence, in expectation our estimator would still be biased (since we would overestimate the ratio and thus underestimate the LENSE).

This argument also seems to be appreciated in statistical arguments/literature (e.g., section 4 in Olkin and Pratt, 1958).

For us it does therefore not appear to be correct to say that NSE (or the LENSE) is an unbiased estimator for a hypothetical true NSE (or the LENSE) — and, in our eyes, the same holds true for the statement made in Lamontagne et al. (2020). Hence, we will keep our formulation as it is since we view it as more elegant and also more similar to the way the NSE is actually used in practice.

Reference: Olkin, I., & Pratt, J. W. (1958). Unbiased estimation of certain correlation coefficients. The annals of mathematical statistics, 201-211.

[L.165] Can it be clarified what the added value is of dividing the MSE of each partition by what's effectively a constant? Is it just that we retain a reference model and therefore the interpretation of NSE > 0?

I don't think that's a bad thing as such, but it would be good to clarify why this special case is more helpful than just looking at the MSE (which, to my understanding, will always be bounded by the MSE values of the partitions).

Yes. This is correct. If one would use the LENSE then its advantage over the MSE would be that it has a similar interpretation of the meaning of 1 and the 0 that the NSE has. We will clarify this in the revised manuscript.

[Figure caption for Fig.4] It might be interesting to show Is = 0 in a specific different color because this could be considered the "intuitive range" of Eq. 2

The argumentation makes sense and we therefore tried to color 0 separately but did not like the visual clutter (see provided example in the attached document). We therefore prefer to keep the figure as is, but will add the resulting figure to Appendix D).

[L. 205ff] I'm not convinced presenting this analysis as CDFs is very meaningful. Even though both metrics have the same range <-inf, 1], the interpretation of each within that range, I assume, is not necessarily the same.

While we agree that the interpretation is not necessarily the same (what remains is the meaning of 1 as perfect predictions and 0 as the point where the simulation is as good an estimator for the evaluation set as the mean is an estimator for the reference set), we believe the CDF nicely illustrates the offset that is caused by the change in measures.

One aspect that is also lost this way is the per-case comparison: what is the relation between LENSE and NSE for the collection of individual cases? I could be interesting to add a scatter plot of NSE vs LENSE to see if anything meaningful appears.

We agree with the observation that the individual case relation is lost. We will add a scatter plot (see attached document) and discuss it in the revised manuscript.

[L.230ff] This is essentially the same recommendation as Schaefli & Gupta (2007), as well as others, have made. I think your work gives a mathematical explanation for this, but also highlights a related but different problem: that NSE and similar scores

cannot easily be compared between different periods for the same basin. I think this is a key contribution that deserves mention here and in the abstract, because the consequences are quite far-reaching.

We agree and will mention it here and in the abstract.

[L.255f] I appreciate that the code is available. I noticed in the online readme that the simulations themselves will be made available after peer review. I think that's a good idea because it's a bit cumbersome having to run the full training procedure to get a few hydrographs to work with, and I couldn't manage to get this done within a reasonable time frame.

That said, the results here are not conditional on how the simulations were generated. Some fiddling around with my own data did not provide any evidence that would contrast the statements in this paper.

Thank you. We completely agree with this assessment.

Couple of code comments:

run-grid-experiment.py

- l142, l156. Is the "eps" inside the brackets strictly necessary? It seems to me that the one outside the brackets would prevent any divide-by-zero errors on its own, whereas the one inside the brackets increases the denominator by a bit but doesn't do much else.

No, it is not necessary. It just a remainder comes from the paranoid stance of the first author while doing experiments. As stated in the comment it does not have an influence on the results.

run-lense-experiment.py

- l88. Is this unit conversion step correct? Unless I'm mistaken somewhere, the first coefficient should be 28.316846592, not 28316846.592

The conversion should be correct. The factor 28.316846592 would convert from feet to liters, but we convert to mm.